# Clinical Spectrum and CSF Findings in Patients with West-Nile Virus Infection, a Retrospective Cohort Review

**DOI:** 10.3390/diagnostics12040805

**Published:** 2022-03-25

**Authors:** Imre Bakos, Mohamed Mahdi, László Kardos, Anna Nagy, István Várkonyi

**Affiliations:** 1Infectology Clinic, University of Debrecen Clinical Center, 4031 Debrecen, Hungary; dr.bakos.imre@kenezy.unideb.hu (I.B.); mohamed@med.unideb.hu (M.M.); dr.kardos.laszlo@kenezy.unideb.hu (L.K.); 2National Reference Laboratory for Viral Zoonoses, National Public Health Center, 1097 Budapest, Hungary; nagy.anna@oki.antsz.hu

**Keywords:** West Nile virus, infection, CSF findings, case review

## Abstract

West Nile Virus (WNV) infection is a world-wide zoonotic disease transmitted by mosquitoes. The infection is usually self-limiting; however, elderly patients or those with comorbidities are predisposed to developing severe, and sometimes fatal complications of the infection. Recently, the incidence of WNV infection in Europe had seen a sharp increase, as compared to previous years. We are currently reporting on the clinical presentation and laboratory findings in 23 cases of WNV infection, of which one resulted in a fatal outcome. The clinical picture was predominantly that of meningitis/meningoencephalitis of varying severity. One patient suffered a fatal outcome, and a rare manifestation of chorioretinal lesions and iridocyclitis was also reported as a result of WNV infection. Cerebrospinal fluid analysis predominantly showed lymphocytic pleocytosis, and total protein levels were increased in all but three of the patients. Levels of total protein in CSF samples were found to show a positive correlation with age. Given the ever-increasing incidence of WNV infection in Europe, a high index of clinical suspicion should always accompany cases of meningitis, especially during the summer period, as a similar epidemic pattern is predicted to recur.

## 1. Introduction

The West Nile virus (WNV) is a member of the Flavivirus genus, a single-stranded RNA virus measuring around 50 nm in diameter. It was isolated for the first time from a woman with fever in Uganda in 1937 [1]; not long thereafter, it was implicated in several outbreaks of febrile illnesses and neuroinvasive disease in the Mediterranean basin in the period between the 1950s and 1960s [2,3,4,5]. Humans are incidental and “dead end” hosts, as they are unable to support the threshold level of viremia required for onward transmission of the virus to the mosquito vector, while wild birds are considered the optimal, most important hosts to support viral replication. It is worth mentioning that many vertebrate species are also susceptible to the infection, although; as in the case with humans, the low level of viremia sustained in such hosts is not thought to be sufficient for viral transmission [6,7]. Transmission of infection is vectored by the *Aedes* spp., *Culex* spp., or *Anopheles* spp. mosquitoes [8]; however, transmission by blood transfusions, vertical transmission via the placenta and breast milk, and organ transplantation, have all been documented [9,10,11]. The clinical spectrum of West Nile infection is wide, varying from an asymptomatic infection to flaccid paralysis and fatal neuroinfection, with the risk of fatal outcome being highest in the elderly, infants, and in immunocompromised patients [12,13].

Given the commonly benign course of WNV infection, most cases go unnoticed. According to the European Center for Disease Prevention and Control (ECDC), a spike in confirmed infections in 2018 had far exceeded that of previous years, with a total of 1503 cases of human WNV infections reported in the EU. In 2021, 139 cases of WNV infections were reported [14], a sharp decrease since 2018. In Hungary, the National Public Health Center (NPHC) has declared 215 autochthonous, and 10 imported cases of WNV infections in 2018, with Hajdú-Bihar County reporting the highest number of cases [14,15]. This county is located in eastern Hungary, with a population of around 530,000 according to Hungarian central statistical office (2018), bordering Romania from the east, and Szabolcs-Szatmár-Bereg, Borsod-Abaúj-Zemplén, Jász-Nagykun-Szolnok and Békés Counties from the north, west, and south. In this paper, we are reporting on the clinical spectrum and CSF laboratory findings in patients diagnosed with WNV infection admitted during summer period of 2018 to the Infectology clinic in at the University of Debrecen Clinical Center, in Hajdú-Bihar County, Hungary. In contrast to the previous years, and the years thereafter, 2018 had the highest rate of WNV infections.

## 2. Materials and Methods

In Hungary, human WNV infections are monitored by the Department of Communicable Diseases Epidemiology and Infection Control (NPHC), Budapest, Hungary, which maintains surveillance records of cases and updates the database regularly. The Infectology ward located in Kenézy Gyula Hospital (University of Debrecen Clinical Center) is a major regional center for infectious diseases in eastern Hungary. In addition to providing clinical services to the local population, patients with infectious diseases in the neighboring counties are also referred to the department if deemed necessary. The 2018/945/EU Implementing Decision of the European Parliament and of the Council laid down the clinical and laboratory case definitions for WNV infection [16]. Data were collected from patient admission records retrospectively, during the period of 1 July–30 September 2018, only confirmed cases of WNV infection were included in the analysis, supported by laboratory tests. Parameters included in the analysis were demographics, age, sex, duration of hospital stay, presence of comorbidities, contact with animals or pets, travel history, treatment and complications. The laboratory analysis included complete blood count (CBC), liver and kidney function tests, serum ion and glucose levels, serological and molecular tests, and cerebrospinal fluid (CSF) analysis from lumbar punctures. Where necessary, interdepartmental consultations were carried out.

### 2.1. Serological Testing

WNV specific microbiological investigation was performed at the National Reference Laboratory for Viral Zoonoses of the NPHC; Budapest, Hungary. First round antibody screening was carried out with WNV indirect immunofluorescence assay (IFA) developed in-house [17]. Serum and CSF samples were tested for the presence of anti-WNV IgG, IgM, and IgA antibodies. To exclude possible serological cross-reactions, subsequent IFA antibody testing was carried out for other endemic, human pathogenic flaviviruses; such as Tick-borne encephalitis virus (TBEV) and Usutu virus (USUV). For detection of anti-WNV IgM antibodies, IgM Capture ELISA test (Focus Diagnostics, DiaSorin Molecular LLC, Cypress, CA, USA) was also used. Test procedure was carried out as described in the manufacturer’s instruction. All serological results were interpreted according to the patient’s flavivirus vaccination status and travel history. For confirmation of IFA and ELISA results, WNV and USUV serum neutralization assay was also performed in agreement with the European Union’s laboratory case definition criteria for confirmation of acute WNV infection [16].

### 2.2. Molecular Biological Analysis

EDTA-treated whole blood and urine samples were used for PCR diagnostics. WNV lineage 1 and lineage 2 specific real-time reverse-transcription polymerase chain reaction (RT-PCR) was performed [18]. Each positive reaction was confirmed with WNV-specific nested RT-PCR assay [19], and PCR amplicons were sequenced by Sanger sequencing for lineage identification. Viral sequences were submitted to the GenBank database (https://www.ncbi.nlm.nih.gov/pubmed/, accessed on 22 March 2022). Besides serology, it is advised to exclude possible USUV infection using molecular methods, due to the close serological relatedness of WNV and USUV. Therefore, in each case of a negative WNV PCR test result, samples were further examined for the presence of USUV RNA by real-time RT-PCR method [20].

### 2.3. Statistical Analysis

Variables were described using standard statistics. Unadjusted comparisons of sample subgroups in terms of categorical outcomes were based on Fisher’s exact test; for categorical outcomes, Student’s two-sample *t* tests were used if the corresponding distributional assumptions were satisfied, and Wilcoxon’s rank-sum tests otherwise. Continuous variables were inspected for distribution shape and transformed if this improved normality. Associations between continuous variables were analyzed using linear regression. Relationship curvature was assessed and a quadratic explanatory term was used when it resulted in a better fit. Adjustment for age and sex was used unless doing so meant no appreciable contribution to the model. Model plausibility was visualized by superimposing fitted values, their 95% confidence interval, and an empirical trendline obtained by locally weighted linear smoothing, over the scatter plot of the outcome versus the key explanatory variable. Statistical evaluation was restricted to serologically confirmed cases. Significance was defined as *p* < 0.05. The statistical package Stata version 15 (StataCorp LLC, College Station, TX, USA) was used for data handling and analysis.

## 3. Results

### 3.1. Demographics and Comorbidities

During the analysis period, a total of 26 patients with suspected WNV infection were admitted, out of which 23 were confirmed by serological and/or PCR-based techniques. Of the 23 confirmed cases, 12 were males and 11 females, and the average age was 57.9 years. Presentation was that of meningitis or meningoencephalitis in patients referred from regional general practitioners (GPs) or other nearby hospitals.

Most of the patients had preexisting medical conditions, with hypertension and diabetes mellitus (type I and II) being the most commonly found comorbidities, present in 43% and 26% of patients, respectively. Other reported significant comorbidities were: ischemic heart disease, asthma, alcoholism, renal and prostatic tumors, and Crohn’s disease. 

History of contact with family pets (dogs, cats, poultry, swine, rodents, sheep, and goats) was reported in 11 (52%) of the patients, and only 3 had a positive history of contact with birds and horses.

### 3.2. Presenting Symptoms

The most common presenting symptoms were that of meningitis/meningoencephalitis; such as headache and fever of 38 °C or more (16 patients), vomiting and dizziness (12 patients), generalized weakness (10 patients), bradykinesia (3 patients) and photophobia (2 patients). One patient (X-08) suffered status epilepticus on the second day of admission, and eventually required intubation and mechanical ventilation. This patient was treated with Clonazepam and Levetiracetam, and showed a spontaneous recovery.

Neurological symptoms also included impaired hearing and lower right-sided extremity paresis in one patient (Y-10), and a left-sided hemiparesis in another (Y-08). 

Other nonspecific and uncommonly reported symptoms of WNV infection were also present in some patients; such as a nonpruritic maculo-papular rash involving the upper and lower extremities in one patient, muscle and joint pain, bilateral tinnitus, and iridocyclitis.

### 3.3. CSF Analysis and Serology

Lumbar CSF analysis revealed pleocytosis (>10 cells per microliter) in all of the patients, with the lowest and highest cell number in the liquor being 16 and 769 cells per microliter, respectively. Lymphocytes were the predominant cell type in CSF samples. When statistical analysis was carried out, no significant difference was found in cell number or cell type predominance when we compared samples from male and female patients (*p* = 0.47, and 0.1, respectively). Cell number and type did not vary with age either (*p* = 0.42, 0.48, respectively), data are illustrated in Figure 1 and Figure 2.

All samples showed an increased total protein level, ranging from 548 mg/L to 1800 mg/L. There was a statistically significant association between older age and a rising total protein level in our samples. Samples from male patients had a significantly higher level of proteins (Figure 3).

Age-adjusted comparison revealed that this was fully due to confounding, explained by male patients being significantly older on average (64.6 years versus 50.5 in females; *p* = 0.02).

The ratio of CSF glucose to serum glucose ranged from 0.42–0.77. However, no association was observed between the ratio of CSF to serum glucose and age or gender (*p* values = 0.96 and 0.18, respectively) (Figure 4).

WNV specific serological tests and RT based PCR to detect viral RNA were carried out on all samples from patients with suspected meningitis/meningoencephalitis. CSF IgM positivity confirmed the acute WNV infection in 10 patients. RT-PCR was positive in 10 patients.

A summary of the WNV microbiological and CSF findings and patient population is provided in Table 1.

### 3.4. Treatment

Following admission with symptoms of meningitis/meningoencephalitis, symptomatic treatment was initiated with Mannitol and Furosemide adjusted for body weight according to hospital protocols, with the exception of a 62-year-old female patient who presented with only mild/moderate symptoms and did not require treatment (X-03). In the case of 6 patients, due to the severity of the symptoms and an unclear clinical picture, Dexamethasone and an antibiotic was also initiated (Ceftriaxone and/or Moxifloxacin).

The mean duration of in-hospital stay was 7.1 days (range: 4–13 days), after which the patients showed a spontaneous recovery and improvement of laboratory values, thereafter discharged with a follow-up appointment for assessment.

### 3.5. Complications

While the majority of patients showed a spontaneous recovery, some suffered complications. A 64-year-old male patient (Y-09) with a history of hypertension and alcoholism, developed bilateral pulmonary emboli, as was evident by the lower segmental changes in both lobes on thoracic CT examination. This complication developed on the 7th day of admission, and reportedly manifested on the 11th day after symptoms first started. 

A 77-year-old patient (Y-03) was admitted with symptoms of meningitis, fever and bronchitis. On the 5th day of admission, a chest X-ray showed a 3 cm right-sided perihilar pulmonary infiltrate, and the patient was started on antibiotic therapy (Moxifloxacin).

The only fatal outcome we observed during this period was in the case of an 89-year-old patient (Y-02). His previous history was nonsignificant, apart from a cholecystectomy, and minor urinary tract infections. On admission, heteroanamnesis revealed that weakness, productive cough, nausea, loss of appetite, and fever were the reason for seeking medical attention. His blood work showed decreased kidney function, increased liver enzymes, and thrombocytopenia. CSF analysis revealed pleocytosis (830/µL) and an increased protein level (1017 mg/L). On the 7th day of admission, a 2.5 cm right-sided perihilar pulmonary infiltrate was detected by chest X-ray. The patient was commenced on antibiotic therapy (Ceftriaxone). Despite therapy, his clinical status did not regress, confusion and high fever persisted. On the 9th day of admission, the patient died as a result of cardiorespiratory insufficiency. 

## 4. Discussion

In 2018, there was a sharp rise in the incidence of WNV infection during the period of July to September in Debrecen, Hungary, from which most of the cases were reported. This was in line with the trend observed in Europe at the time, where higher case numbers of infections were reported as compared to the previous year [14]. A total of 225 cases of WNV infection were reported in Hungary, with the majority of cases occurring in the Hajdú-Bihar County (32 human cases), compared to only 7 cases in the previous year [21]. During the observed period, of the 32 reported cases, 26 were admitted to our department, out of which 23 were confirmed by serological and/or RT-PCR tests. The majority of patients were residents from the Hajdú-Bihar County, while 6 patients were from the neighboring Jász-Nagykun-Szolnok County, who were referred to our department. In the beginning of the outbreak, WNV infection was suspected in patients presenting with signs and symptoms of meningitis/meningoencephalitis, after exclusion of bacterial and other viral etiologies. Suspicion of infection was then confirmed by serological tests and RT-PCR. Viral nucleic acid amplification revealed WNV lineage 2 circulation, as described in previous transmission seasons in Hungary. 

The patient population was predominantly middle aged, with the youngest patient being a 23-year-old female, while the oldest was an 89-year-old male patient, who unfortunately suffered a fatal outcome as a complication of the infection. This was the only fatal case in our department relating to WNV infection. 

Clinical presentation of WNV was mostly typical of that described in the literature, with symptoms of meningitis/meningoencephalitis that ranged in severity from mild to fatal. Of interest, iridocyclitis presenting in patient X-04—a 51-year-old female diabetic patient—is not a common manifestation of WNV infection. This patient was admitted to our ward with fever, headache and vision disturbance manifesting as deteriorating visual acuity. Ophthalmologic evaluation revealed a long-standing nonproliferative diabetic retinopathy, diabetic maculopathy, and bilateral cataracts, all known previously. Recent findings included left-sided iridocyclitis and chorioretinopathy as detected by Fluorescein Angiography. To our knowledge, in the literature, only two cases reportedly presented with chorioretinal lesions and iridocyclitis as a manifestation of the infection [22], more commonly encountered ocular manifestations being vitreitis, uveitis and chorioretinitis [23,24,25]. 

In conclusion, clinical manifestation of WNV infection is primarily that of self-limiting meningitis/meningoencephalitis. Laboratory findings; especially CSF analysis, were mostly consistent with those found in other viral causes of meningitis. It is worth noting though, as revealed by our statistical analysis, that total CSF protein level was found to increase with age, and also was significantly higher in male patients as compared to females, although this was probably due to confounding. Taking all this into consideration, the mostly self-limiting clinical course and noncharacteristic laboratory findings in WNV infection still pose a clinical challenge in case suspicion and early detection, especially in areas where the infection has been relatively uncommon, which undoubtedly affects the proper reporting of actual incidence and prevalence in many European countries. It is therefore of vital importance to include WNV as part of the routine serology panel in patients presenting with meningitis in Europe, especially during the summer periods.

## Figures and Tables

**Figure 1 diagnostics-12-00805-f001:**
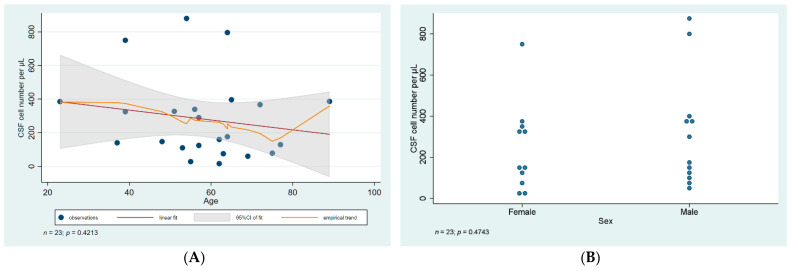
Statistical analysis of CSF cell number. (**A**) Scatter plot of CSF cell number against age with linear regression fit and empirical trend superimposed. (**B**) Categorical scatter plot of CSF cell number by sex. CSF—cerebrospinal fluid; CI—confidence interval.

**Figure 2 diagnostics-12-00805-f002:**
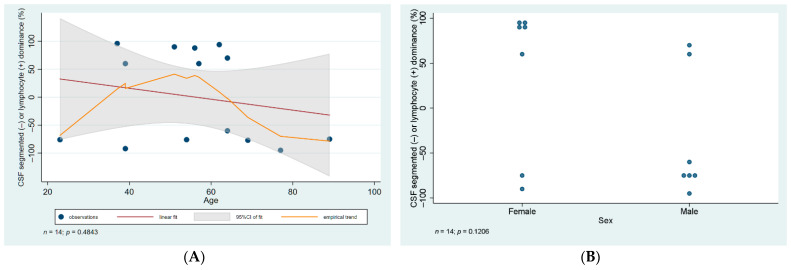
Statistical analysis of CSF cell type. (**A**) Scatter plot of CSF cell type against age with linear regression fit and empirical trend superimposed. (**B**) Categorical scatter plot of CSF cell type by sex. CSF—cerebrospinal fluid; CI—confidence interval. Data of cell type analysis was not available in the case of 9 patients.

**Figure 3 diagnostics-12-00805-f003:**
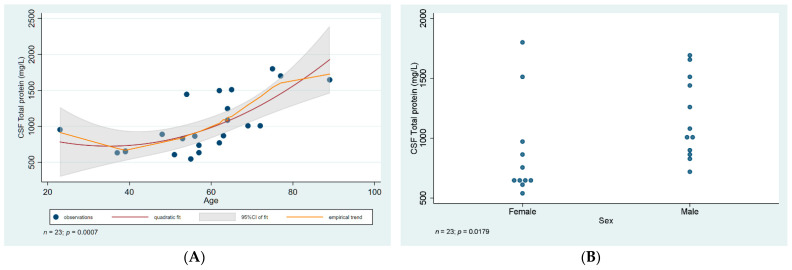
Statistical analysis of CSF total protein level. (**A**) Scatter plot of CSF total protein level against age with linear regression fit and empirical trend superimposed. (**B**) Categorical scatter plot of CSF total protein level by sex. CSF—cerebrospinal fluid; CI—confidence interval.

**Figure 4 diagnostics-12-00805-f004:**
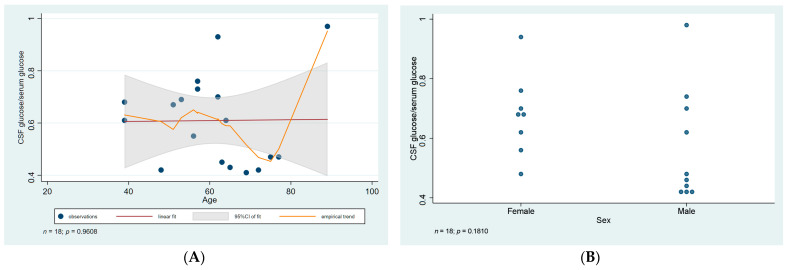
Statistical analysis of CSF glucose to serum glucose ratio. (**A**) Scatter plot of CSF/serum glucose ratio against age with linear regression fit and empirical trend superimposed. (**B**) Categorical scatter plot of the ratio by sex. CSF—cerebrospinal fluid; CI—confidence interval. Data was not available in the case of 5 patients.

**Table 1 diagnostics-12-00805-t001:** Case index and summary of CSF findings and serology results. Single asterisk (*) denotes anti-WNV IgG seroconversion was detected in paired sera, while double asterisk (**) denote probable acute WNV infection in accordance with the clinical picture. IFA—indirect immunofluorescence assay, NA—not applicable, ND—not determined. Indeterminate WNV real-time RT-PCR result was determined by a cycle threshold value >40.00. Presentation was considered as: “mild” if symptoms were limited to flu-like, fever, headache, body aches and rash; “moderate” if mild symptoms in addition to significant impairment of CNS functions, muscle tone were present; or “severe” if any of the above in addition to altered mental status, pneumonia, status epilepticus, hemiparesis, or pulmonary embolism occurred.

ID	Age	Sex	CSF WNV IgM IFA	Serum WNV IgM ELISA	WNV RT-PCR Blood	WNV RT-PCR Urine	CSF Cell Number Per µL	CSF Cell Type	CSF Total Protein (mg/L)	Ratio of CSF to Serum Glucose	Treatment Duration (Days)	Presentation
**X01**	56	F	Positive	Positive	Negative	Negative	339	88% lymphocytes	864	0.55	9	mild
**X04 ***	51	F	Negative	Positive	Negative	Negative	327	90% lymphocytes	607	0.67	6	moderate
**X03**	62	F	Negative	Positive	Positive	Positive	16	ND	771	0.7	4	moderate
**X02 ***	39	F	Negative	Positive	Negative	Negative	750	92% segmented	657	0.61	7	mild
**X08**	62	F	Positive	Positive	Negative	Negative	160	94% lymphocytes	1497	0.93	10	severe
**X06**	23	F	Positive	Positive	Positive	Positive	385	76% segmented	955	ND	4	moderate
**X05 ****	75	F	NA	Positive	Negative	Negative	78	ND	1800	0.47	6	moderate
**X07**	55	F	Positive	Positive	Negative	Negative	28	ND	548	ND	7	mild
**X09**	37	F	Positive	Positive	NA	Negative	140	96% lymphocytes	633	ND	6	moderate
**X13 ***	39	F	Indeterminate	Positive	Indeterminate	Negative	325	60% lymphocytes	649	0.68	6	moderate
**X11**	57	F	Positive	Positive	Indeterminate	Negative	124	ND	635	0.76	4	severe
**Y01**	69	M	NA	Positive	Positive	Negative	60	77% segmented	1008	0.41	6	severe
**Y02**	89	M	Indeterminate	Positive	Positive	Positive	386	75% segmented	1647	0.97	9	severe/fatal
**Y03 ****	77	M	NA	Positive	Negative	Positive	129	95% segmented	1701	0.47	13	severe
**Y04**	65	M	Positive	Positive	Positive	Positive	396	ND	1510	0.43	5	moderate
**Y06**	53	M	Positive	Positive	Positive	Indeterminate	110	ND	830	0.69	8	moderate
**Y07**	72	M	NA	Positive	Positive	Indeterminate	367	ND	1008	0.42	7	moderate
**Y05 ***	48	M	Negative	Positive	Negative	Positive	147	ND	890	0.42	7	moderate
**Y08**	54	M	Positive	Positive	Positive	Positive	880	76% segmented	1445	ND	8	severe
**Y09 ****	64	M	Negative	Positive	Negative	Negative	796	60% segmented	1086	0.61	8	severe
**Y10 ****	64	M	NA	Positive	Negative	NA	176	70% lymphocytes	1246	ND	5	severe
**Y11 ****	63	M	NA	Positive	Indeterminate	Negative	75	ND	870	0.45	11	mild
**Y12**	57	M	Positive	Positive	Negative	Negative	290	60% lymphocytes	737	0.73	7	mild

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
