# Peer review of "Clinical Spectrum and CSF Findings in Patients with West-Nile Virus Infection, a Retrospective Cohort Review"

_diagnostics, 2022, doi:10.3390/diagnostics12040805_

Round 1

Reviewer 1 Report

I would like to thank the Editor for giving me the opportunity to read the manuscript "Clinical spectrum and CSF findings in patients with West-Nile virus infection, a retrospective cohort review".

This is a very clear report regarding clinical characteristics of a population diagnosed with West Nile virus meningitis in Hungary.

Despite this condition has already been reported and investigated from multiple points of view, I believe that the present report must be praised to contribute to the awareness on this contition and to illustrate laboratory results in a homogeneous population of patients.

I would reccommend publication after minor spell checks throughout the manuscript.

Author Response

Thank you indeed for taking the time and effort reviewing our manuscript, we really appreciate it. We have corrected some minor spelling mistakes and modified the table and some text. Thank you again.

Reviewer 2 Report

Manuscript ID: diagnostics-1619154

Title: Clinical spectrum and CSF findings in patients with West-Nile virus infection, a retrospective cohort review

The manuscript describes the clinical presentations and CSF findings of 23 cases of West Nile infections. The topic is of great importance in Europe, since the number of cases is on the rise, but the infection is still not that common in most parts, to be easily recognized by clinicians. The manuscript would benefit greatly from a more detailed description of results of microbiological investigations and better organization of the manuscript. The manuscript shows great potential and adds additional knowledge to the diagnostics of WNV infections. Nevertheless, some changes would improve the manuscript.

Comments:

  1. Materials and methods, Serological testing, line 89: Please update reference 16 for WNV case definition (Decision 2018/945)
  2. Figure 1-4: The legend font is too small, please correct.
  3. Results: The CSF findings are well described. Is it possible to indicate on what day of the illness the CSF samples were taken? Were CSF samples taken only when neurological symptoms were present or at the beginning of hospitalization? Though the sample is relatively small, some statistical analysis based on the clinical presentation (for example moderate vs. severe illness) would be interesting to see.
  4. Results: The microbiological results are too unspecific. Is the presence of IgM, presented in Table 1 determined in CSF as required by the Laboratory criteria for case conformation? Please specify in either the text or Table 1. The authors’ state in the materials and methods they performed the neutralization assay, but the results are not presented. Especially for patients X-12 and X-13 it would be interesting to see the results.
  5. Results: please state what type of sample was positive in case of positive RT-PCR (CSF, blood, urine).
  6. Table 1: Please add the clinical presentation or at least the form of the disease (mild, moderate, severe). The descriptions of clinical presentations are scattered all through the manuscript, but it would be clearer and more informative, if the data are added to the Table 1.
  7. The references have double numbering.

Author Response

Thank you for your comments and time. We will hereby try to address your concerns to the best of our ability.

Comments:

  • Materials and methods, Serological testing, line 89: Please update reference 16 for WNV case definition (Decision 2018/945)

Thank you, reference updated

  • Figure 1-4: The legend font is too small, please correct.

The font has now been increased to 10 pts.

  • Results: The CSF findings are well described. Is it possible to indicate on what day of the illness the CSF samples were taken? Were CSF samples taken only when neurological symptoms were present or at the beginning of hospitalization? Though the sample is relatively small, some statistical analysis based on the clinical presentation (for example moderate vs. severe illness) would be interesting to see.
  • Table 1: Please add the clinical presentation or at least the form of the disease (mild, moderate, severe). The descriptions of clinical presentations are scattered all through the manuscript, but it would be clearer and more informative, if the data are added to the Table 1.

Thank you for your comment. The lumbar puncture was performed when neurological symptoms manifested and clinical suspicion arose. All the patients who were referred to us had some sort of neurological manifestation,  and they were all hospitalized, so the samples were taken in the beginning of admission. Unfortunately, it would be very difficult to do the requested statistical analysis at this time, especially given the small sample number.

As all the patients were admitted, in addition to the fact that there is no standardized scale for classification of the severity of WNV infection, it was quite difficult to categorize severity of presentation for the studied population, however, as requested, we have added a very broad definition of severity to the table and complemented it with the following paragraph:

Presentation was considered as “mild” if: symptoms were limited to flu-like, fever, headache, body aches and rash, “moderate”: mild symptoms in addition to significant impairment of CNS functions, muscle tone, or “severe”: if any of the above in addition to altered mental status, pneumonia, status epilepticus, hemiparesis, or pulmonary embolism occurred.

  • Results: The microbiological results are too unspecific. Is the presence of IgM, presented in Table 1 determined in CSF as required by the Laboratory criteria for case conformation? Please specify in either the text or Table 1. The authors’ state in the materials and methods they performed the neutralization assay, but the results are not presented. Especially for patients X-12 and X-13 it would be interesting to see the results.

We have now complemented our table with detailed description of neutralization assays and serology.

  • Results: please state what type of sample was positive in case of positive RT-PCR (CSF, blood, urine).

The table has now been updated. Blood and urine samples were tested by WNV specific real-time RT PCR, while viral nucleic acid detection was not applied for CSF samples. WNV RNA is undetectable in CFS, in most of the cases. A PCR negative result in CSF does not rule out the possibility of an acute WNV neuroinfection.

Reference:

Pacenti, M., Sinigaglia, A., Franchin, E., Pagni, S., Lavezzo, E., Montarsi, F., Capelli, G., & Barzon, L. (2020). Human West Nile virus lineage 2 infection: Epidemiological, clinical, and virological findings. Viruses, 12(4). https://doi.org/10.3390/v12040458

  • The references have double numbering.

Double numbering has been resolved.